# Added value of anxiolytic benzodiazepines in predictive models on severe delirium in patients with acute decompensated heart failure: A retrospective analysis

Kei Kawada[1,2]*, Hitoshi Fukuda[3], Toru Kubo[4], Tsuyoshi Ohta[5], Tomoaki Ishida[2], Shumpei Morisawa[1,2], Tetushi Kawazoe[2], Manami Okamoto[2], Hiroko Fujita[2], Kohei Jobu[2], Yasuyo Morita[2], Tetsuya Ueba[3], Hiroaki Kitaoka[4], Mitsuhiko Miyamura[1,2]

1 Graduate School of Integrated Arts and Sciences Kochi University, Oko town, Nankoku City, Kochi, Japan, 2 Department of Pharmacy, Kochi Medical School Hospital, Oko town, Nankoku City, Kochi, Japan, 3 Department of Neurosurgery, Kochi Medical School Kochi University, Oko town, Nankoku City, Kochi, Japan, 4 Department of Cardiology and Geriatrics, Kochi Medical School Kochi University, Oko town, Nankoku City, Kochi, Japan, 5 Department of Neurosurgery, National Cerebral and Cardiovascular Center Hospital, Suita City, Osaka, Japan

◉ These authors contributed equally to this work.
* jm-kei_kawada@kochi-u.ac.jp

## Abstract

### Background

Delirium in patients with acute decompensated heart failure (ADHF) is associated with poor clinical outcomes. Although some medications have been reported as risk factors for delirium, their impact on patients with ADHF is still unclear. This study aimed to determine the association of specific medication use with delirium and their additive predictive value in models based on conventional risk factors.

### Methods and results

In this single-center, retrospective study, 650 patients treated for ADHF were included. Fifty-nine patients (9.1%) had delirium. In multivariate analysis, anxiolytic benzodiazepines [odds ratio (OR): 6.4, 95% confidence interval (CI): 2.8–15], mechanical ventilation or noninvasive positive pressure ventilation (OR: 6.0, 95% CI: 2.9–12), depression (OR: 3.2, 95% CI: 1.5–6.5), intensive care or high care unit admission (OR: 2.9, 95% CI: 1.5–5.6), male sex (OR: 2.0, 95% CI: 1–3.7), and age (OR: 1.1, 95% CI: 1–1.1) were independently associated with severe delirium. The predictive model that included anxiolytic benzodiazepines had a significantly better discriminatory ability for the incidence of severe delirium than the conventional model.

### Conclusions

The use of anxiolytic benzodiazepines was independently correlated with severe delirium, and their use in models based on conventional risk factors had an additive value for predicting delirium in patients with ADHF.

**Data Availability Statement:** All relevant data are within the manuscript and its Supporting Information files.

**Funding:** The authors received no specific funding for this work.

**Competing interests:** The authors have declared that no competing interests exist.

## Introduction

Delirium is a common symptom in patients with acute decompensated heart failure (ADHF), and it is associated with prolonged hospitalization and increased morbidity and mortality [1,2]. Specific patient characteristics and severity of heart failure (HF) have been posited as risk factors for delirium in patients with ADHF [1–3]. Although many medications have been associated with the development of delirium in surgical or intensive care unit (ICU) patients [4–6], only a few studies have demonstrated an association between the type of medication and the development of delirium in hospitalized patients with ADHF. Therefore, this study aimed to investigate the association between the use of specific medications and the development of delirium in patients with ADHF and to assess the discriminative performance of a novel outcome prediction model comprising conventional risk factors and medication use.

## Materials and methods

This study was approved by the Ethics Committee on Medical Research of Kochi Medical School (31–126). The procedures used in this study adhered to the tenets of the Declaration of Helsinki. The requirement for informed consent was waived, given the retrospective nature of the study. The confidentiality and anonymity of patient data were maintained throughout.

### Patient selection and study design

This study was a retrospective cohort study of consecutive patients with ADHF hospitalized for more than 3 days between January 2015 and December 2019 at a single center. ADHF, diagnosed by a cardiologist, was defined as new-onset decompensated HF or decompensation of chronic HF with symptoms that warrant hospitalization according to the guidelines [7]. All patients had clinical and radiographic evidence of pulmonary congestion [8]. Patients who were already delirious at admission were excluded. The patients' baseline characteristics were all considered independent risk factors for delirium [1,2,5,6,9–13]. They included age, sex, body mass index (BMI), New York Heart Association class, and ICU or high care unit (HCU) admission. Treatment-related factors (mechanical ventilation or noninvasive positive pressure ventilation [NPPV], percutaneous coronary intervention, and cardiac surgery); laboratory data on admission (brain natriuretic peptide, aspartate aminotransferase, alanine aminotransferase, γ-glutamyltranspeptidase, and estimated glomerular filtration rate); comorbidities (hypertension, dyslipidemia, hyperuricemia, insomnia, diabetes, cerebrovascular disease, depression, chronic obstructive pulmonary disease, anxiety disorder, and dementia); habits (current smoking and habitual drinking); and the use of medications for the treatment of HF and its complications, were analyzed for any correlation with the occurrence of delirium within 7 days of admission.

Delirium was diagnosed according to the Confusion Assessment Method for the ICU, a standardized evidence-based tool that enables the quick and accurate identification of delirium in both clinical and research settings [14]. Subsequently, the severity of delirium was classified according to the Richmond Agitation Sedation Scale, and grade 2 or higher was diagnosed as severe [15–17], which required the administration of antipsychotics or sedatives.

### Medications used

We investigated the medications used for the treatment of HF and its complications, including those that had been used before hospitalization and continued after admission and those that were prescribed within 3 days of admission. Medications used for the treatment of HF included angiotensin-converting enzyme inhibitors or angiotensin receptor blockers (ACE-

inhibitor/ARB), beta-blockers, diuretics, digoxin, and pimobendan. Medications for the treatment of complications included antiarrhythmics, anticoagulants, antihistamines, benzodiazepine receptor agonists for insomnia, anxiolytic benzodiazepines, asthma medication, antiepileptics, steroids, selective serotonin reuptake inhibitors, and serotonin noradrenaline reuptake inhibitors. Anxiolytic benzodiazepines included etizolam and clotiazepam. Benzodiazepine receptor agonists for insomnia included zolpidem, eszopiclone, brotizolam, triazolam, and estazolam.

## Statistical analysis

Categorical variables were compared using Fisher's exact test, and ratios or ordinal variables were compared using Student's t-test. Results are expressed as mean ± standard deviation for continuous variables and number (percentage) for categorical variables. The odds ratios (ORs) and 95% confidence intervals (CIs) were also determined using logistic regression analysis. $P < 0.05$ was considered statistically significant for all tests. All P-values $< 0.001$ are expressed as $P < 0.001$.

First, the association of all variables with the occurrence of severe delirium was evaluated using univariate analysis. Multivariate logistic regression models were created using backward stepwise selection to include variables with P values $< 0.1$ in the univariate analysis to exclude confounding factors and test for an independent association with the occurrence of severe delirium.

We also assessed whether including or excluding medications associated with severe delirium from models adjusted for traditional risk factors would lead to a greater ability to predict the occurrence of delirium.

C statistics analogous to the area under the receiver operating curve were estimated to compare the discriminatory abilities of these models [18]. The statistical significance of differences was compared using the method described by DeLong et al [19]. The increase in discriminatory ability after the inclusion of medications that correlated with severe delirium was further examined by assessing the net reclassification improvement (NRI) and integrated discrimination improvement (IDI) [20]. To create cross-tabulation, we classified the probability of severe delirium into four categories, $< 2.9\%$, 2.9% to 5.6%, 5.6% to 10%, and $> 10\%$, based on the median value and the quartiles of the predicted probability in all patients. The continuous NRI values evaluated changes in the estimated prediction probabilities between different models. The IDI considered differences in discrimination slopes between different models, where the discrimination slope is defined as the difference between the mean of the estimated prediction probabilities, taken as continuous variables for individuals with events and the corresponding mean for those without events. Statistical analyses were conducted using R software (version 3.3.1, R Foundation for Statistical Computing; http://www.Rproject.org).

## Results

### Patient characteristics and medication use

During the study period, 659 patients required hospitalization for ADHF and were assessed for eligibility. Seven patients were hospitalized for less than 72 hours, and 2 were delirious at admission. Therefore, 9 patients were excluded, and 650 patients with ADHF were finally included in the analysis. The baseline characteristics of the patients on admission are shown in Table 1. The cohort comprised 366 men and 284 women with a mean age of 75.2 ± 12.7 years.

During the study period, 59 patients (9.1%) experienced severe delirium. Patients with delirium had longer hospital stays than those without delirium (47.0 ± 81.6 vs. 25.7 ± 21.0 days, $P < 0.001$). The medications used for the treatment of HF and its complications are

**Table 1. Baseline characteristics of patients.**

| Variable | Delirium | No delirium | *P*-value |
|---|---|---|---|
| No. of patients | 59 | 591 | |
| Age (years), mean ± SD | 81.7 ± 9.1 | 74.5 ± 12.8 | < 0.001 |
| Sex, male, n (%) | 36 (61.0) | 330 (55.8) | 0.49 |
| BMI (kg/m$^2$), mean ± SD | 22.0 ± 3.6 | 23.2 ± 4.7 | 0.041 |
| NYHA class III or IV on admission, n (%) | 49 (83.1) | 433 (73.2) | 0.12 |
| ICU or HCU admission, n (%) | 43 (72.9) | 276 (46.7) | < 0.001 |
| Length of hospital stay (days), mean ± SD | 47.0 ± 81.6 | 25.7 ± 21.0 | < 0.001 |
| Treatment during hospitalization, n (%) | | | |
| Mechanical ventilation or NPPV | 19 (32.2) | 36 (6.1) | < 0.001 |
| PCI | 1 (1.7) | 17 (2.8) | 0.99 |
| Cardiac surgery | 1 (1.7) | 33 (5.5) | 0.35 |
| Laboratory data on admission | | | |
| BNP (pg/mL), mean ± SD | 1,037.4 ± 796.5 | 920.0 ± 1,047.2 | 0.4 |
| AST (U/L), mean ± SD | 63.7 ± 116.9 | 74.2 ± 405.4 | 0.84 |
| ALT (U/L), mean ± SD | 43.6 ± 78.9 | 50.4 ± 197.5 | 0.79 |
| γGTP (U/L), mean ± SD | 57.6 ± 59.0 | 70.7 ± 72.0 | 0.18 |
| eGFR (mL/min/1.73 m$^2$), mean ± SD | 38.4 ± 17.8 | 47.3 ± 23.9 | 0.0053 |
| Medical history, n (%) | | | |
| Hypertension | 48 (81.3) | 459 (77.7) | 0.62 |
| Dyslipidemia | 35 (59.3) | 341 (57.7) | 0.89 |
| Hyperuricemia | 24 (40.7) | 279 (47.2) | 0.41 |
| Insomnia | 29 (49.2) | 231 (39.1) | 0.16 |
| Diabetes | 16 (27.1) | 185 (31.3) | 0.56 |
| Cerebrovascular disease | 10 (16.9) | 110 (18.6) | 0.86 |
| Depression | 17 (28.8) | 67 (11.3) | < 0.001 |
| COPD | 3 (5.1) | 29 (4.8) | 0.99 |
| Anxiety disorder | 6 (10) | 39 (6.5) | 0.28 |
| Dementia | 8 (13.6) | 15 (2.5) | < 0.001 |
| Habits, n (%) | | | |
| Current smoking | 7 (11.9) | 75 (12.7) | 0.99 |
| Habitual drinking | 14 (23.7) | 148 (25.0) | 0.99 |

ALT, alanine transaminase; AST, aspartate aminotransferase; BMI, body mass index; BNP, brain natriuretic peptide; COPD, chronic obstructive pulmonary disease; eGFR, estimated glomerular filtration rate; HCU, high care unit; ICU, intensive care unit; NYHA, New York Heart Association; NPPV, noninvasive positive pressure ventilation; PCI, percutaneous coronary intervention; SD, standard deviation; γ-GTP, γ-glutamyltransferase.

summarized in Table 2. Regarding medications for HF, 628 patients (96.6%), 447 patients (68.8%), and 423 patients (65.1%) were prescribed diuretics, beta-blockers, and ACE-inhibitors/ARBs, respectively. For the treatment of complications, 43 patients (6.6%) were prescribed anxiolytic benzodiazepines.

## Risk factors for delirium in patients with ADHF

In univariate analysis (Tables 1 and 2), delirium was found to be associated with older age, lower BMI, greater rate of ICU or HCU admission, greater rate of mechanical ventilation or NPPV, lower estimated glomerular filtration rate, depression, dementia, and use of anxiolytic benzodiazepines. As shown in Table 3, multivariate analysis indicated that anxiolytic benzodiazepines (OR: 6.4, 95% CI: 2.8–15, P < 0.001), mechanical ventilation or NPPV (OR: 6.0, 95%

**Table 2. Medications used for the treatment of HF and its complications.**

| Variable | Delirium | No delirium | *P*-value |
|---|---|---|---|
| **Medications for the treatment of HF, n (%)** | | | |
| Diuretic | 57 (96.6) | 571 (96.6) | 0.99 |
| Beta-blocker | 42 (71.2) | 405 (68.5) | 0.77 |
| ACE-inhibitor/ARB | 39 (66.1) | 384 (65.0) | 0.99 |
| Pimobendan | 12 (20.3) | 76 (12.9) | 0.11 |
| Digoxin | 3 (5.1) | 21 (3.6) | 0.47 |
| **Medications for the treatment of complications, n (%)** | | | |
| Anticoagulant | 53 (89.8) | 535 (90.5) | 0.83 |
| Antiarrhythmic | 21 (35.6) | 277 (46.9) | 0.1 |
| Benzodiazepine receptor agonists for insomnia | 15 (25.4) | 127 (21.5) | 0.51 |
| Antihistamine | 7 (11.9) | 76 (12.9) | 0.99 |
| Anxiolytic benzodiazepines | 14 (23.7) | 29 (4.8) | < 0.001 |
| Steroids | 3 (5.1) | 35 (5.9) | 0.99 |
| Asthma medicine | 4 (6.8) | 33 (5.5) | 0.77 |
| Antiepileptic | 2 (3.4) | 13 (2.2) | 0.64 |
| SSRI/SNRI | 1 (1.7) | 5 (0.9) | 0.44 |

ACE-inhibitor, angiotensin-converting enzyme inhibitor; ARB, angiotensin II receptor blocker; HF, heart failure; SSRI, selective serotonin reuptake inhibitor; SNRI, serotonin and noradrenaline reuptake inhibitor.

CI: 2.9–12, P < 0.001), depression (OR: 3.2, 95% CI: 1.5–6.5, P = 0.0021), ICU or HCU admission (OR: 2.9, 95% CI: 1.5.5.6, P = 0.002), male sex (OR: 2.0, 95% CI: 1–3.7, P = 0.042), and age (OR: 1.1, 95% CI: 1–1.1, P < 0.001) were independent predictors of severe delirium.

## Predictive models for delirium

To evaluate the impact of anxiolytic benzodiazepines on the incidence of severe delirium, we compared the discriminatory abilities of models with and without anxiolytic benzodiazepines. The C statistic for a model created with mechanical ventilation or NPPV, depression, ICU or HCU admission, male sex, and increasing age was 0.81, which was significantly increased by the addition of anxiolytic benzodiazepines (C statistic = 0.86, P = 0.037). The cross-tabulation for functional outcome between models with and without anxiolytic benzodiazepines is shown in Table 4. Among the ADHF patients who developed severe delirium, using the model with anxiolytic benzodiazepines improved classification in five patients and worsened classification in three patients. The use of this model also improved classification in 153 patients who did

**Table 3. Multivariate analysis of factors that are predictive of delirium.**

| Variable | OR | 95% CI | *P*-value |
|---|---|---|---|
| Anxiolytic benzodiazepines | 6.4 | 2.8–15 | < 0.001 |
| Mechanical ventilation or NPPV | 6.0 | 2.9–12 | < 0.001 |
| Depression | 3.2 | 1.5–6.5 | 0.0021 |
| ICU or HCU admission | 2.9 | 1.5–5.6 | 0.002 |
| Male sex | 2.0 | 1–3.7 | 0.042 |
| Age | 1.1 | 1–1.1 | < 0.001 |

CI, confidence interval; HCU, high care unit; ICU, intensive care unit; NPPV, noninvasive positive pressure ventilation; OR, odds ratio.

**Table 4. Reclassification for the risk of delirium in patients with ADHF.**

| Basic model | Basic model + anxiolytic benzodiazepines | | | | |
|---|---|---|---|---|---|
| | < 2.9% | 2.9–5.6% | 5.6–10% | > 10% | Total |
| **Patients who had severe delirium** | | | | | |
| < 2.9% | 1 | 0 | 1 | 0 | 2 |
| 2.9–5.6% | 0 | 6 | 0 | 2 | 8 |
| 5.6–10% | 0 | 1 | 6 | 2 | 9 |
| > 10% | 0 | 0 | 2 | 38 | 40 |
| Total | 1 | 7 | 9 | 42 | 59 |
| **Patients who did not have severe delirium** | | | | | |
| < 2.9% | 156 | 1 | 1 | 1 | 159 |
| 2.9–5.6% | 51 | 90 | 0 | 15 | 156 |
| 5.6–10% | 0 | 73 | 74 | 6 | 153 |
| > 10% | 0 | 0 | 29 | 94 | 123 |
| Total | 207 | 163 | 103 | 116 | 591 |

ADHF, acute decompensated heart failure.

not experience severe delirium while worsening classification in 24 patients. The NRI and IDI were calculated to be 0.34 (P < 0.001) and 0.045 (P < 0.001), respectively, suggesting that the inclusion of anxiolytic benzodiazepines conferred added value to predictive models of severe delirium based on conventional risk factors.

## Discussion

In this study, we showed that the use of anxiolytic benzodiazepines was associated with severe delirium in patients with ADHF. In addition, the use of anxiolytic benzodiazepines augmented the ability to discriminate severe delirium for a predictive model based on conventional risk factors.

Delirium is an acute change in cognition with altered consciousness and impaired attention that fluctuates over time [21]. The prevalence of delirium is 15% on admission [22], and this condition is associated with poor clinical outcomes [23–29]. The incidence of delirium is higher in elderly patients, patients with emergency hospital admission, and patients requiring invasive treatments [23,30–32]. Moreover, delirium is observed in approximately 35% of hospitalized patients with HF [1,2]. In the present study, severe delirium (grade ≥ 2 using the Richmond Agitation Sedation Scale) was found in 9.1% of hospitalized ADHF patients. Patients with HF comprise a high-risk group for delirium because they are normally elderly, have a high emergency hospitalization rate, and require highly invasive treatments such as respiratory management and cardiac catheterization [1,2] In patients with HF, delirium is also associated with a longer hospital stay, readmission for worsening HF, and higher short-term mortality rates [1,2]. Therefore, it is important to clarify the risk factors for delirium in patients with HF.

In this study, anxiolytic benzodiazepines were significantly associated with severe delirium in patients with HF. Patients with HF are more likely to experience anxiety disorders or insomnia, which necessitates treatment with anxiolytic benzodiazepines or benzodiazepine receptor agonists, respectively [33,34]. Drug-induced delirium is thought to result from transient thalamic dysfunction caused by exposure to medications that interfere with central glutamatergic, γ-aminobutyric acidergic, dopaminergic, and cholinergic pathways at critical sites of action [35]. The pharmacological effects of benzodiazepine agonists may cause delirium because they

reduce the corticostriatal glutamatergic tone and induce transient thalamic filtering dysfunction [35]. In this study, more specifically, anxiolytic benzodiazepines for anxiety disorder induced delirium while benzodiazepine receptor agonists for insomnia did not. A possible explanation is the distinct duration of the drug effect in a day; anxiolytic benzodiazepines are taken two or three times a day while benzodiazepine receptor agonists are taken only once before sleep. The stable diurnal plasma drug concentration of anxiolytic benzodiazepines may also have continuous negative consequences, including daytime sedation, ataxia, anterograde memory disturbance, dependence, and more importantly, delirium [36–39]. Considering that anxiety disorders themselves have been reported as a risk factor for delirium, it is also possible that delirium is induced by the deterioration of anxiety disorders in patients taking anxiolytic benzodiazepines because of acute changes in their HF status or emergency admission [40,41].

It is difficult to predict the incidence of delirium in patients with ADHF. Although advanced age and a history of cerebrovascular disease have been reported as risk factors for delirium in patients with ADHF [1–3], some patients still present with delirium in the absence of these conventional risk factors. Because approximately 80% of patients with ADHF are elderly, have comorbidities, and are often prescribed several medications [42,43], we assumed that some medications in this subpopulation might mediate the incidence of delirium by influencing the central nervous system [35]. In this study, we showed that the use of anxiolytic benzodiazepines augmented the discriminatory ability of our predictive model based on conventional risk factors such as advanced age and comorbidities; this suggested that the use of anxiolytic benzodiazepines had an additive predictive value distinct from that of conventional risk factors. Given that conventional risk factors for delirium mainly comprise physical and environmental burdens, including mechanical ventilation and ICU, the use of anxiolytic benzodiazepines may increase the likelihood of delirium through increased psychiatric burden.

Our study had some limitations. Given the retrospective design and single-center setting, the findings should be interpreted with caution. In addition, potential interactions among variables may not have been completely excluded by the multivariable analyses. In this study, we included potential risk factors for delirium as covariates during the regression analysis. However, delirium is more common in the elderly and their precipitating factors for ADHF are distinct from those in younger patients. Thus, future studies should focus on elucidating whether elderly-specific precipitating factors for ADHF, including infection and arrhythmia [44], could also be risk factors for delirium. Furthermore, the patients' pre-admission cognitive function and functional status, which might be associated with delirium, could not be evaluated because of missing medical record information. Therefore, confirmation of the associations between these factors and delirium requires prospective studies. However, our results suggest that the use of anxiolytic benzodiazepines, which we found as a new risk factor for delirium, could be a therapeutic strategy for the prevention of delirium in patients with HF, considering that medication is modifiable as opposed to age, severity of HF, and comorbidities.

In conclusion, anxiolytic benzodiazepine use may be independently associated with delirium in patients with ADHF. Thus, incorporating the use of anxiolytic benzodiazepines into a predictive model based on conventional risk factors may offer additional predictive value. Further investigations are warranted to determine whether avoiding the use of anxiolytic benzodiazepines reduces the incidence of delirium in patients with HF.

## Supporting information

**S1 Table. Complete patient dataset.**
(XLS)

## Author Contributions

**Conceptualization:** Kei Kawada, Hitoshi Fukuda.

**Data curation:** Kei Kawada, Tomoaki Ishida, Shumpei Morisawa, Tetushi Kawazoe, Manami Okamoto, Hiroko Fujita.

**Formal analysis:** Kei Kawada, Hitoshi Fukuda.

**Funding acquisition:** Kohei Jobu, Mitsuhiko Miyamura.

**Investigation:** Kei Kawada, Tomoaki Ishida, Shumpei Morisawa, Tetushi Kawazoe, Manami Okamoto, Hiroko Fujita.

**Methodology:** Kei Kawada, Hitoshi Fukuda.

**Supervision:** Toru Kubo, Tsuyoshi Ohta, Tetsuya Ueba, Hiroaki Kitaoka, Mitsuhiko Miyamura.

**Writing – original draft:** Kei Kawada.

**Writing – review & editing:** Hitoshi Fukuda, Toru Kubo, Tsuyoshi Ohta, Kohei Jobu, Yasuyo Morita, Tetsuya Ueba, Hiroaki Kitaoka, Mitsuhiko Miyamura.

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
