## [Decision Letter · Decision Letter 0]

25 Jan 2021

PONE-D-20-40765

Anxiolytic benzodiazepines may induce severe delirium in patients with acute decompensated heart failure

PLOS ONE

Dear Dr. KAWADA,

Thank you for submitting your manuscript to PLOS ONE. After careful consideration, we feel that it has merit but does not fully meet PLOS ONE’s publication criteria as it currently stands. Therefore, we invite you to submit a revised version of the manuscript that addresses the points raised during the review process.

We look forward to receiving your revised manuscript.

Kind regards,

Pasquale Abete

Academic Editor

PLOS ONE

Journal Requirements:

2. Please modify the title to ensure that it is meeting PLOS’ guidelines (https://journals.plos.org/plosone/s/submission-guidelines#loc-title). In particular, the title should be "specific, descriptive, concise, and comprehensible to readers outside the field", and in this case the title may be misleadingly implying that your study is an experimental one, rather than an observational one.

Reviewers' comments:

Reviewer's Responses to Questions

**Comments to the Author**

1. Is the manuscript technically sound, and do the data support the conclusions?

Reviewer #1: Yes

Reviewer #2: Yes

2. Has the statistical analysis been performed appropriately and rigorously? 

Reviewer #1: Yes

Reviewer #2: Yes

3. Have the authors made all data underlying the findings in their manuscript fully available?

Reviewer #1: Yes

Reviewer #2: No

4. Is the manuscript presented in an intelligible fashion and written in standard English?

Reviewer #1: Yes

Reviewer #2: Yes

5. Review Comments to the Author

Reviewer #1: This study is aimed to evaluate the association of anxiolytic benzodiazepines use with delirium and their additive predictive value in models based on conventional risk factors. Data derived from a single-center, retrospective study, that enrolled 650 patients treated for ADHF. Fifty-nine patients (9.1%) had delirium. In multivariate analysis, anxiolytic benzodiazepines, mechanical ventilation or noninvasive positive pressure ventilation, depression, intensive care or high care unit admission, male sex, and age were independently associated with severe delirium. The predictive model that included anxiolytic benzodiazepines had a significantly better discriminatory ability for the incidence of

severe delirium than the conventional model.

The study is of great interest, even if data derived from a single-center study, results confirmed also in HF patients the use of anxiolytic benzodiazepines as risk factor for severe delirium. Table 1 and 2 should be better presented.

Reviewer #2: The authors retrospectively studied if delirium in patients with acute decompensated heart failure (ADHF) was associated with poor clinical outcomes in 650 patients treated for ADHF. Fifty-nine patients (9.1%) had delirium. In multivariate analysis, anxiolytic benzodiazepines [odds ratio (OR): 6.4, 95% confidence interval (CI): 2.8–15], mechanical ventilation or noninvasive positive pressure ventilation (OR: 6.0, 95% CI: 2.9–12), depression (OR: 3.2, 95% CI: 1.5–6.5), intensive care or high care unit admission (OR: 2.9, 95% CI: 1.5–5.6), male sex (OR: 2.0, 95% CI: 1–3.7), and age (OR: 1.1, 95% CI: 1–1.1) were independently associated with severe delirium. The predictive model that included anxiolytic benzodiazepines had a significantly better discriminatory ability for the incidence of severe delirium than the conventional model.

The manuscript is interesting and topic. However, I have some concerns about the precipitating factors for ADHF considered. Please see and discuss Testa G et al. Precipitating factors in younger and older adults with decompensated chronic heart failure: are they different? J Am Geriatr Soc. 2013 Oct;61(10):1827-8.

6. PLOS authors have the option to publish the peer review history of their article (what does this mean?). If published, this will include your full peer review and any attached files.

Reviewer #1: **Yes: **Francesco Cacciatore

Reviewer #2: No

---

## [Author Response · Author response to Decision Letter 0]

5 Feb 2021

February 4, 2021

Professor Pasquale Abete

Academic Editor

PLOS ONE

Dear Prof. Abete:

We appreciate the opportunity to resubmit our article titled “Anxiolytic benzodiazepines may induce severe delirium in patients with acute decompensated heart failure”, with manuscript ID PONE-D-20-40765, for publication in PLOS ONE as an Original Contribution. 

We believe we have responded to all the comments from the Academic Editor and the Reviewer carefully, with corresponding changes made directly to the manuscript where appropriate. The revised portions of the manuscript are provided as Tracked Changes for your convenience. 

Accompanying this cover letter, please find a revised version of our manuscript and a rebuttal letter. 

Thank you for your kind consideration.

Sincerely,

Kei Kawada

Department of Pharmacy, Kochi Medical Hospital School 

185-1 Kohasu, Oko town, Nankoku City, Kochi, Japan

Tel:088-866-5811 

Fax:088-880-2456 

Email: jm-kei_kawada@kochi-u.ac.jp

Journal comments:

Comment 1: Please ensure that your manuscript meets PLOS ONE's style requirements, including those for file naming. The PLOS ONE style templates can be found at

Response: Thank you for this comment. We have ensured that all PLOS ONE style requirements have been followed, including those for file naming.

Comment 2: Please modify the title to ensure that it is meeting PLOS’ guidelines (https://journals.plos.org/plosone/s/submission-guidelines#loc-title). In particular, the title should be "specific, descriptive, concise, and comprehensible to readers outside the field", and in this case the title may be misleadingly implying that your study is an experimental one, rather than an observational one.

Response: Thank you for this comment. We have modified the title to “Added value of anxiolytic benzodiazepines in predictive models on severe delirium in patients with acute decompensated heart failure: a retrospective analysis” accordingly.

Comment 3: We note that you have indicated that data from this study are available upon request. PLOS only allows data to be available upon request if there are legal or ethical restrictions on sharing data publicly. For information on unacceptable data access restrictions, please see http://journals.plos.org/plosone/s/data-availability#loc-unacceptable-data-access-restrictions.

Response: Thank you for this comment. We have now obtained permission from the Kochi Medical Hospital School’s institutional review board to publish all the study data. Therefore, we have provided the data as Supplementary Information files and changed the data availability statement.

Reviewers' comments:

Response: Thank you for your positive comments. We have thoroughly reviewed the whole manuscript and responded to all the comments carefully. Our point-by-point responses are listed below. The revised portions of the manuscript are provided as Tracked Changes.

Review Comments to the Author:

Reviewer #1: This study is aimed to evaluate the association of anxiolytic benzodiazepines use with delirium and their additive predictive value in models based on conventional risk factors. Data derived from a single-center, retrospective study, that enrolled 650 patients treated for ADHF. Fifty-nine patients (9.1%) had delirium. In multivariate analysis, anxiolytic benzodiazepines, mechanical ventilation or noninvasive positive pressure ventilation, depression, intensive care or high care unit admission, male sex, and age were independently associated with severe delirium. The predictive model that included anxiolytic benzodiazepines had a significantly better discriminatory ability for the incidence of severe delirium than the conventional model.

The study is of great interest, even if data derived from a single-center study, results confirmed also in HF patients the use of anxiolytic benzodiazepines as risk factor for severe delirium. Table 1 and 2 should be better presented.

Response: Thank you for your positive feedback and comment. It would appear that Tables 1 and 2 become misaligned when the document is converted to PDF at the time of submission. We have ensured that this mistake is not repeated when resubmitting our manuscript.

Reviewer #2: The authors retrospectively studied if delirium in patients with acute decompensated heart failure (ADHF) was associated with poor clinical outcomes in 650 patients treated for ADHF. Fifty-nine patients (9.1%) had delirium. In multivariate analysis, anxiolytic benzodiazepines [odds ratio (OR): 6.4, 95% confidence interval (CI): 2.8–15], mechanical ventilation or noninvasive positive pressure ventilation (OR: 6.0, 95% CI: 2.9–12), depression (OR: 3.2, 95% CI: 1.5–6.5), intensive care or high care unit admission (OR: 2.9, 95% CI: 1.5–5.6), male sex (OR: 2.0, 95% CI: 1–3.7), and age (OR: 1.1, 95% CI: 1–1.1) were independently associated with severe delirium. The predictive model that included anxiolytic benzodiazepines had a significantly better discriminatory ability for the incidence of severe delirium than the conventional model.

The manuscript is interesting and topic. However, I have some concerns about the precipitating factors for ADHF considered. Please see and discuss Testa G et al. Precipitating factors in younger and older adults with decompensated chronic heart failure: are they different? J Am Geriatr Soc. 2013 Oct;61(10):1827-8.

Response:

Thank you for your positive feedback and suggestion. We have added discussion points on the precipitating factors for ADHF and referenced Testa G et al. Precipitating factors in younger and older adults with decompensated chronic heart failure: are they different? J Am Geriatr Soc. 2013 Oct;61(10):1827-8 accordingly. The limitation below about the differ in younger and older adults was added in the discussion section and references.

Discussion (page 17, lines 220)

“In this study, we included potential risk factors for delirium as covariates during the regression analysis. However, delirium is more common in the elderly and their precipitating factors for ADHF are distinct from those in younger patients. Thus, future studies should focus on elucidating whether elderly-specific precipitating factors for ADHF, including infection and arrhythmia [44], could also be risk factors for delirium.”

References (page 24, lines 342)

(44) Gianluca T, David DM, Francesco C, Gaetano, Daniele D'A, Gianluigi G, et al. Precipitating factors in younger and older adults with decompensated chronic heart failure: are they different?. J Am Geriatr Soc. 2013;61: 1827-1828.”

---

## [Decision Letter · Decision Letter 1]

6 Apr 2021

Added value of anxiolytic benzodiazepines in predictive models on severe delirium in patients with acute decompensated heart failure: a retrospective analysis

PONE-D-20-40765R1

Dear Dr. KAWADA,

We’re pleased to inform you that your manuscript has been judged scientifically suitable for publication and will be formally accepted for publication once it meets all outstanding technical requirements.

Kind regards,

Pasquale Abete

Academic Editor

PLOS ONE

Additional Editor Comments (optional):

No further comments.

Reviewers' comments:

Reviewer's Responses to Questions

**Comments to the Author**

1. If the authors have adequately addressed your comments raised in a previous round of review and you feel that this manuscript is now acceptable for publication, you may indicate that here to bypass the “Comments to the Author” section, enter your conflict of interest statement in the “Confidential to Editor” section, and submit your "Accept" recommendation.

Reviewer #1: All comments have been addressed

Reviewer #2: All comments have been addressed

2. Is the manuscript technically sound, and do the data support the conclusions?

Reviewer #1: Yes

Reviewer #2: Yes

3. Has the statistical analysis been performed appropriately and rigorously? 

Reviewer #1: Yes

Reviewer #2: Yes

4. Have the authors made all data underlying the findings in their manuscript fully available?

Reviewer #1: Yes

Reviewer #2: Yes

5. Is the manuscript presented in an intelligible fashion and written in standard English?

Reviewer #1: Yes

Reviewer #2: Yes

6. Review Comments to the Author

Reviewer #1: (No Response)

Reviewer #2: The manuscript is really improved and all the questions are correctly answered. The manuscript merits to be published in PONE.

7. PLOS authors have the option to publish the peer review history of their article (what does this mean?). If published, this will include your full peer review and any attached files.

Reviewer #1: **Yes: **francesco cacciatore

Reviewer #2: No

---

## [Editor Report · Acceptance letter]

12 Apr 2021

PONE-D-20-40765R1 

Added value of anxiolytic benzodiazepines in predictive models on severe delirium in patients with acute decompensated heart failure: a retrospective analysis 

Dear Dr. Kawada:

I'm pleased to inform you that your manuscript has been deemed suitable for publication in PLOS ONE. Congratulations! Your manuscript is now with our production department. 

Kind regards, 

on behalf of

Prof. Pasquale Abete 

Academic Editor

PLOS ONE